# Neuroprotective Action of Tacrolimus before and after Onset of Neonatal Hypoxic–Ischaemic Brain Injury in Rats

**DOI:** 10.3390/cells12222659

**Published:** 2023-11-20

**Authors:** Madeleine J. Smith, Tayla Penny, Yen Pham, Amy E. Sutherland, Graham Jenkin, Michael C. Fahey, Madison C. B. Paton, Megan Finch-Edmondson, Suzanne L. Miller, Courtney A. McDonald

**Affiliations:** 1The Ritchie Centre, Hudson Institute of Medical Research, Clayton, VIC 3168, Australia; madeleine.smith@monash.edu (M.J.S.); tayla.penny@hudson.org.au (T.P.); yennie.pham@hudson.org.au (Y.P.); amy.sutherland@hudson.org.au (A.E.S.); graham.jenkin@monash.edu (G.J.); michael.fahey@monash.edu (M.C.F.); suzie.miller@monash.edu (S.L.M.); 2Department of Obstetrics and Gynaecology, Monash University, Clayton, VIC 3168, Australia; 3Department of Paediatrics, Monash University, Clayton, VIC 3168, Australia; 4Cerebral Palsy Alliance Research Institute, Speciality of Child and Adolescent Health, Sydney Medical School, Faculty of Medicine and Health, The University of Sydney, Sydney, NSW 2050, Australia; madison.paton@cerebralpalsy.org.au (M.C.B.P.); mfinch-edmondson@cerebralpalsy.org.au (M.F.-E.)

**Keywords:** immune system, immunosuppression, neonatal brain injury, neuroprotection, T-cells

## Abstract

(1) Background: Neonatal brain injury can lead to permanent neurodevelopmental impairments. Notably, suppressing inflammatory pathways may reduce damage. To determine the role of neuroinflammation in the progression of neonatal brain injury, we investigated the effect of treating neonatal rat pups with the immunosuppressant tacrolimus at two time points: before and after hypoxic–ischaemic (HI)-induced injury. (2) Methods: To induce HI injury, postnatal day (PND) 10 rat pups underwent single carotid artery ligation followed by hypoxia (8% oxygen, 90 min). Pups received daily tacrolimus (or a vehicle) starting either 3 days before HI on PND 7 (pre-HI), or 12 h after HI (post-HI). Four doses were tested: 0.025, 0.05, 0.1 or 0.25 mg/kg/day. Pups were euthanised at PND 17 or PND 50. (3) Results: All tacrolimus doses administered pre-HI significantly reduced brain infarct size and neuronal loss, increased the number of resting microglia and reduced cellular apoptosis (*p* < 0.05 compared to control). In contrast, only the highest dose of tacrolimus administered post-HI (0.25 mg/kg/day) reduced brain infarct size (*p* < 0.05). All doses of tacrolimus reduced pup weight compared to the controls. (4) Conclusions: Tacrolimus administration 3 days pre-HI was neuroprotective, likely mediated through neuroinflammatory and cell death pathways. Tacrolimus post-HI may have limited capacity to reduce brain injury, with higher doses increasing rat pup mortality. This work highlights the benefits of targeting neuroinflammation during the acute injurious period. More specific targeting of neuroinflammation, e.g., via T-cells, warrants further investigation.

## 1. Introduction

Neonatal brain injury is a leading cause of morbidity and mortality and can lead to life-long conditions including cerebral palsy and epilepsy. Causes of neonatal brain injury include infection, hypoxia–ischaemia (HI) and perinatal stroke [1]. Additionally, antenatal compromise, such as in utero infection, chronic hypoxia and foetal growth restriction, make the neonatal brain more susceptible to injury [2,3,4,5]. Neuroinflammation plays a key role in the pathogenesis of neonatal brain injury across these conditions. Specifically, immune cells, including neutrophils, microglia, macrophages, mast cells and astrocytes, become activated. Sustained innate immune cell activation can have detrimental effects on healthy brain tissue and cause damage to neurons and oligodendrocyte precursors [6].

Although the adaptive immune response in neonates was previously considered underdeveloped compared to adults, there is mounting evidence regarding the contribution of the peripheral immune system and T-cells in neonatal brain injury [7]. T-cells are derived from the thymus and play a key role in cell-mediated immunity. Upon antigen binding to the major histocompatibility (MHC) complex, T-cells become activated and may secrete a variety of pro- and anti-inflammatory cytokines that assist with cell death and clearance. T-cells have been shown to infiltrate the brain following neonatal injury [8,9,10], and lymphocyte-deficient mice exposed to a preterm HI insult were protected against the development of white matter injury [11]. Unlike adult T-cells, neonatal CD4+ T-cells are biased towards the production of T helper type 2 (Th2) cytokines, including interleukin (IL)-4, IL-5 and IL-13 [12]. Although the role of T-cells in neonatal brain injury remains largely unknown, modulating the T-cell response following neonatal injury may be beneficial. 

Tacrolimus (FK506) is an immunosuppressive drug that reduces T-cell activity and is commonly used to prevent the rejection of solid organ transplants in humans [13]. Tacrolimus’ mechanism of action includes binding to immunophilin FK506-binding protein-12, and interfering with T-cell calcineurin/NFAT signalling, which, in turn, suppresses T-cell activity [14]. Tacrolimus has also been shown to be neuroprotective in preclinical adult models of peripheral nerve damage [15], ischaemic brain injury [16] and spinal cord injury [17]. Studies investigating the potential of tacrolimus for treating neonatal brain injury are limited; however, one study has shown that tacrolimus administration causes a dose-dependent reduction in the number of macrophages within the injured neonatal brain [18]. Administering tacrolimus in a model of neonatal HI brain injury may allow for characterisation of the immunological mechanisms driving injury, particularly those of the systemic T-cell response. Hence, the first aim of this study was to determine the dose effect of tacrolimus administered pre-HI, to ascertain the effect of suppressing the immune system on the initiation and development of injury. The second aim of this study was to determine the potential of administering tacrolimus post-HI brain injury as an immunomodulatory treatment strategy to reduce neonatal brain injury.

## 2. Materials and Methods

### 2.1. Ethics Approval

All experiments were performed with Animal Ethics approval from Monash Medical Centre Animal Ethics Committee A (MMCA/2019/28) and were performed in accordance with the Australian National Health and Medical Research Council guidelines for the care and use of animals for scientific purposes.

### 2.2. Animal Handling

Time-mated pregnant Sprague Dawley rat dams were sourced from the Monash Animal Research Platform and transported to the Monash Medical Centre Animal Facility one week prior to birth. They were housed in individual boxes in standard housing conditions in rooms with a 12 h light/dark cycle. The dams were allowed to birth naturally and were only disturbed to count the pups on postnatal day (PND) 2–3. Within each litter, pups were randomly assigned to experimental groups, based on surgery times and sex, to control for any neuroprotective effects associated with prolonged isoflurane exposure during anaesthesia or the effect that sex may have on injury or neuroprotection, respectively. Pups were weighed and health-checked daily.

### 2.3. Experimental Groups

The overall aim of this study was to examine the effect of tacrolimus administration in an HI model of neonatal brain injury, but first, the short-term effect of tacrolimus on the systematic cytokine response and the long-term tolerability of tacrolimus were determined. Consequently, three separate experimental cohort groups were studied (animal numbers in Appendix A).

Group 1: First, we aimed to determine the effect of tacrolimus on the systemic cytokine response. Pups were administered tacrolimus (Sigma-Aldrich PHR1809) at a dose of 0.025, 0.05, 0.1, 0.25, 0.5 or 1.0 mg/kg/day in 200 μL phosphate-buffered saline (PBS) or PBS alone (200 μL) via the intraperitoneal (IP) route starting on PND 7 until post-mortem on PND 11. Animals were culled via IP administration of an overdose of sodium pentobarbitone (0.1 mg/g Virbac Pty Ltd., Milperra, Australia).

Group 2: To assess the long-term tolerability of tacrolimus, pups were administered tacrolimus at a dose of 0.025, 0.05, 0.1, 0.25, 0.5 or 1.0 mg/kg/day of PBS alone starting on PND 7 until PND 49. Animals were euthanised on PND 50 via lethal inhalation of carbon dioxide followed by decapitation.

Group 3: To assess the effect of tacrolimus on HI brain injury (the main aim of this study), tacrolimus (0.025, 0.05, 0.1 or 0.25 mg/kg/day) or PBS was administered daily, starting on either PND 8 (3 days pre-HI) or PND 11 (12 h post-HI). Sham control pups underwent sham surgery and received daily PBS injections. Post-mortem was conducted on PND 17. Animals were culled via IP administration of an overdose of sodium pentobarbitone (0.1 mg/g Virbac Pty Ltd., Milperra, Australia).

In all experimental groups, tacrolimus was administered within a two-hour window in the morning.

### 2.4. Animal Surgery

The pups in experimental Group 3 underwent HI injury, induced on PND 10 via permanent unilateral carotid artery ligation, followed by exposure to hypoxia as previously published [10]. The pups were separated from their mother and placed on a 37 °C heat pad before and after surgery. Pups were initially anaesthetised via inhalation of 4% isoflurane and maintained at 2% for the duration of the surgery. A small midline incision was made in the neck, and the left carotid artery was exteriorised before occluding with an electrocautery device (Bovie, Clearwater, FL, USA). The wound was sutured using a 6–0 polypropylene suture, and isoflurane inhalation was stopped. Bupivacaine (Aspen Pharmacare, St Leonards, NSW, Australia) was applied to the surgery site for pain management. The pups were returned to their dam for a 1 h recovery period. Following this, the pups that underwent carotid artery ligation were placed in a humidified hypoxic chamber for 90 min at 35–36 °C and 8% oxygen, balanced with 92% nitrogen. Control sham pups underwent sham surgery, where the artery was exteriorised but not ligated. Following surgery, control sham pups were returned to their dam for a 1 h recovery period, and then, placed on a 37 °C heating pad and exposed to room air for 90 min. Following the 90 min hypoxia/control period, all pups were returned to the dam for recovery. 

### 2.5. Post-Mortem and Tissue Processing

To determine the cytokine response following tacrolimus administration (experimental Group 1), spleens were collected post-mortem and placed in cold Roswell Park Memorial Institute (RPMI) media until processing in the lab. All brains were collected and weighed before being fixed in 10% formalin for 3 days; then, they were processed and embedded in paraffin wax. For histological analysis, the embedded brains were sectioned at 6 μm.

### 2.6. Cytokine Analysis

Quantitative analysis of the pro-inflammatory cytokine IL-4 was performed using a rat cytometric bead array flex set (BD Biosciences, Macquarie Park, Australia) as described previously [19]. Briefly, each spleen was passed through a 100 μm filter to obtain single cell suspensions [20]. Using a 24-well plate, 2.5 × 106 splenocytes were cultured for 48 h in complete RPMI medium either alone or supplemented with 800 ng/mL ionomycin and 20 pg/mL PMA (Sigma-Aldrich, Macquarie Park, Australia), following the manufacturer’s instructions. Data were acquired using a FACSCanto II flow cytometer (BD Biosciences, Macquarie Park, Australia) and analysed using FCAP array software (Soft Flow Inc., Burnsville, MN, USA).

### 2.7. Histology

Neuropathology was assessed in experimental Group 3 pups, including the sham group. For all analyses, duplicate brain sections were used and the results were averaged across the two sections. Percentage brain tissue loss was determined via haematoxylin and eosin staining (H&E, Amber Scientific, Midvale, Australia). Images were acquired via Aperio digital scanning (CS2, Leica Biosystems, Wetzlar, Germany), and the volumes of the left (ipsilateral to the injury) and right (contralateral to the injury) hemispheres were measured using Aperio ImageScope (CS2, Leica Biosystems, Germany). For percentage tissue loss calculations, the difference in area between the contralateral and ipsilateral hemispheres over the contralateral hemisphere area was calculated for each individual animal as follows: (area of the contralateral hemisphere − area of the ipsilateral hemisphere)/(area of the contralateral hemisphere).

### 2.8. Immunohistochemistry

For the immunohistochemical analysis of experimental Group 3 brains, duplicate coronal brain sections at approximately −4.2 mm bregma containing the somatosensory cortex and the CA3 region of the hippocampus in the injured ipsilateral (left) hemisphere were analysed (Iba-1, GFAP, TUNNEL, IL-1β and NeuN). Microglia were identified using ionised calcium-binding adapter molecule 1 (Iba-1; 1:1000, Wako Pure Chemical Industries, Ltd., Osaka, Japan), T-cells were identified using CD3e (1:200, Invitrogen, Waltham, MA, USA), neuronal cell counts were assessed using NeuN (1:1000, Millipore, Burlington, MA, USA), astrocytes were assessed using glial fibrillary acidic protein (GFAP; 1:400, Sigma-Aldrich, Castle Hill, NSW, Australia) and interleukin 1-beta (IL-1β) was assessed using IL-1β antibody (1:1500, Abcam, Cambridge, UK). The sections were incubated in primary antibody at 4 °C overnight. The following day, sections were exposed to a biotinylated secondary antibody for one hour (Goat anti-Rabbit IgG, or Goat anti-Mouse IgG; 1:200; Vector Laboratories, Bulingame, CA, USA), followed by exposure to streptavidin-HRP molecules. Staining was visualised using 3,3-diaminobenzidine (DAB; MP Biomedicals, Santa Ana, CA, USA). 

Apoptotic cell death was identified via terminal deoxynucleotidyl transferase dUTP nick end labelling (TUNEL) staining to detect DNA fragmentation according to the manufacturer’s protocol (ApopTag^®^ Peroxidase In Situ Apoptosis Detection Kit, Merck Group, Darmstadt, Germany). Using this procedure, apoptotic nuclei stained dark brown and, for the quantification of apoptotic cells, five fields of view per duplicate were assessed. 

Duplicate slides were imaged at 400× magnification using bright-field microscopy on an Olympus BX-41 microscope (Olympus, Tokyo, Japan), using three fields of view per duplicate per brain region. To quantify the number of CD3-positive cells, the entire left hemisphere was imaged and cells were counted. Cell counts (Iba-1, CD3, NeuN, TUNNEL) and densitometry (GFAP, IL-1β) were performed using Image J (NIH, Bethesda, Maryland, MD, USA) in both the cortex and CA3 regions. Further, cell counts were performed across the entire left hemisphere for CD3-positive staining. Quantification of microglia cell types was based on cell morphology by classifying the microglia as either resting (branching projections protruding from the cell body) or active (no projections seen, cell body rounded). All assessments were blinded using coded slides and images.

### 2.9. Statistics

Results are expressed as the mean ± standard error of the mean (SEM). Statistical analysis was performed using Prism 9.0 (GraphPad Software). Pup weights were compared using two-way ANOVA, and one-way ANOVA was used for all other outcomes. If one-way ANOVA results were found to be significant, Dunnett’s post hoc analysis was used to compare all groups to the HI group. A value of *p* < 0.05 was considered statistically significant.

## 3. Results

### 3.1. Tacrolimus Reduces the Spleen Cytokine Response at High Doses within Three Days of Initiating Treatment

Following the administration of 0.25 mg/kg/day and 1.0 mg/kg/day of tacrolimus, there was a significant decrease in the secretion of IL-4 protein following the stimulation of spleen cells (Figure 1) compared to the PBS controls.

### 3.2. Long-Term Rat Pup Survival and Weight Decreased with Increasing Tacrolimus Concentration

All doses of tacrolimus reduced pup weights at varying points of the experimental period when compared to the PBS controls (Figure 2B). All rat pups survived in the PBS group, and in the tacrolimus groups with dosing of 0.025 mg/kg/day and 0.1 mg/kg/day (Figure 2C). In the other tacrolimus groups, the rat pup survival rates ranged from 40% to 83%, and there was a general trend of reduced survival with increasing tacrolimus dose (Figure 2C). Given the toxic effect that high doses of tacrolimus had on survival and body weight, we also assessed any potential effect tacrolimus alone may have on the brain and cognition. We found no significant differences in brain weight (Appendix A), short-term memory cognition (as measured via novel-object recognition; Appendix A) or neuronal cell count in the hippocampus (Appendix A) following any dose of tacrolimus compared to the PBS controls. Due to the high mortality rate in the animals administered 0.5 and 1.0 mg/kg/day, these doses were not used in subsequent studies.

### 3.3. Tacrolimus Administration Prior to Hypoxic–Ischaemic Brain Injury Is Neuroprotective

#### 3.3.1. Brain Tissue Loss, Neuronal Cell Loss and Apoptosis

Brain injury was confirmed following HI, as evidenced by significant left hemisphere tissue loss in the HI control group compared to sham control animals (*p* < 0.0001; Figure 3B,C). This was prevented by pre-treatment with tacrolimus three days before HI, with brain tissue loss reduced with all four tacrolimus doses, when compared to the HI control group (*p* < 0.05; Figure 3B,C).

Neuronal cell counts (NeuN+) were not different across groups in the CA3 region of the hippocampus (Figure 3D). In the somatosensory cortex, there was a significant decrease in NeuN+ cells in the HI control group compared to sham group (*p* < 0.05; Figure 3E). Tacrolimus at all doses, except 0.1 mg/kg/day, prevented the loss of neurons in the somatosensory cortex, as evidenced by a significant increase in NeuN+ staining between groups pre-treated with tacrolimus, compared to the HI control group (*p* < 0.05; Figure 3E).

There was a significant increase in cells undergoing apoptosis (TUNEL+) in the HI group compared to sham animals in the CA3 region of the hippocampus and cortex (*p* < 0.01, Figure 3F,G). Pre-treatment with tacrolimus significantly decreased the number of cells undergoing apoptosis in both the CA3 region of the hippocampus (Figure 3F) and cortex (Figure 3G), compared to the HI alone group (*p* < 0.05).

#### 3.3.2. Neuroinflammation

Neuroinflammation was quantified by counting the number of microglia (Iba-1+ cells) and assessing cell morphology as either resting or activated (as described above). The number of resting microglia was not significantly different across groups in the CA3 region of the hippocampus (Figure 4A). In the somatosensory cortex, there was a significant decrease in the number of resting microglia in HI control animals compared to the sham group (Figure 4C). Pre-treatment with low-dose tacrolimus (0.025 and 0.05 mg/kg/day) significantly increased the number of resting microglia in the somatosensory cortex compared to the HI control group (*p* < 0.01, *p* < 0.05, respectively; Figure 4C). The number of resting microglia following pre-treatment with the two highest tacrolimus doses was not significantly different compared to the HI control group. Counts for activated microglia were increased in the HI control group compared to sham group in both the CA3 region of the hippocampus (*p* < 0.05, Figure 4B) and somatosensory cortex (*p* < 0.01, Figure 4D). Pre-treatment with tacrolimus at any dose did not significantly decrease the number of activated microglia compared to the HI control group.

To further investigate the effect of tacrolimus on neuroinflammation, astrogliosis was assessed by measuring the density of GFAP staining. There were no significant differences in GFAP density across groups in the hippocampus or somatosensory cortex (Figure 5A,B). Additionally, pro-inflammatory cytokine IL-1 expression was not significantly different across groups in the brain regions examined (Figure 5C,D). The effect of tacrolimus administration prior to HI on the number of T-cells within the brain was assessed by staining for CD3e. A significant increase in T-cells within the brain was observed in the HI control group compared to sham group (Figure 5E), and there was a trend towards a reduction in T-cells following treatment with 0.025 mg/kg/day (*p* = 0.089) and 0.05 mg/kg/day (*p* = 0.069) compared to the HI control group, although this difference was not statistically significant (Figure 5E).

### 3.4. Tacrolimus Administration after Hypoxic–Ischaemic Brain Injury Is Neuroprotective at High Doses

#### 3.4.1. Brain Tissue Loss, Neuronal Cell Loss and Apoptosis

High-dose tacrolimus (0.25 mg/kg/day) commencing 24 h post-HI significantly reduced brain tissue loss (*p* < 0.01; Figure 6B) compared to the HI control group. Post-HI tacrolimus treatment with the three lower doses did not improve brain tissue loss compared to the HI control group (Figure 6B). Neuronal cell counts in both the CA3 region of the hippocampus and somatosensory cortex were reduced in the HI control group compared to the sham controls (*p* < 0.05; Figure 6D,E). All doses of tacrolimus administered post-HI did not improve the number of neurons compared to the HI control group (Figure 6D,E). The number of apoptotic cells was not different across groups in the CA3 region of the hippocampus (Figure 6F). There was a significant increase in cell death observed in the somatosensory cortex in the HI control group compared to the sham group (*p* < 0.05, Figure 6G), and treatments with the two highest doses of tacrolimus (0.1 and 0.25 mg/kg/day) were able to reduce cell death significantly (*p* < 0.05, Figure 6G) compared to the HI control group.

#### 3.4.2. Neuroinflammation

There was no significant difference in the number of resting microglia between all groups in the CA3 region of the hippocampus with post-HI tacrolimus treatment (Figure 7A). A significant decrease in resting microglia was observed in the somatosensory cortex of the HI control group compared to the sham group (*p* < 0.05, Figure 7C), but post-HI treatment with tacrolimus had no effect on resting microglia compared to the HI control group. Additionally, there was no significant difference in the number of activated microglia in the CA3 region of the hippocampus between groups (Figure 7B). There was a significant increase in activated microglia in the somatosensory cortex of the HI control group compared to the sham group (*p* < 0.01, Figure 7D). The administration of tacrolimus at 0.1 mg/kg/day post-HI significantly decreased the number of activated microglia in the somatosensory cortex compared to the HI control group (*p* < 0.05, Figure 7D). 

The evidence of astrogliosis determined based on GFAP density was not significantly different between groups in either the hippocampus or somatosensory cortex (Figure 8A,B). IL-1β density was significantly increased following high-dose tacrolimus administration post-HI (0.25 mg/kg/day) compared to the HI control group in the hippocampus (*p* < 0.001; Figure 8C) and somatosensory cortex (*p* < 0.0001, Figure 8D). Additionally, IL-1β density was significantly increased in the somatosensory cortex following tacrolimus administration at 0.025 mg/kg/day (*p* < 0.01, Figure 8D) and 0.25 mg/kg/day (*p* < 0.01, Figure 8D) compared to the HI control group. A significant increase in T-cells in the brain was observed in the left hemisphere of the HI control group compared to sham group (Figure 8E). Tacrolimus administration at any dose did not affect the number of T-cells present in the brain compared to the HI control group. 

## 4. Discussion

Neonatal HI brain injury can lead to life-long morbidity. Inflammation plays a central role in the progression of brain injury, making immunomodulatory therapies a potential treatment option [6]. In this study, we investigated the effect of tacrolimus, a potent T-cell inhibitor, when administered before and after the induction of HI neonatal brain injury in rat pups. This is the first study to show that pre-treatment with tacrolimus prior to neonatal HI brain injury can prevent brain tissue loss, neuron loss, apoptotic cell death and T-cell brain infiltration and increase resting microglia, even at low doses. This finding suggests that T-cells, and other immune cells affected by tacrolimus, play a pivotal role in the initiation of neonatal brain injury. However, when tacrolimus administration commenced 24 h after HI, only the highest dose of tacrolimus studied demonstrated neuroprotective benefits. These results support that pre-treatment with the T-cell inhibitor tacrolimus prior to neonatal HI is neuroprotective.

Our results showed that tacrolimus administration prior to HI significantly increased the number of resting microglia present in the cortex and reduced the number of T-cells infiltrating into the brain following HI injury. However, we did not observe a significant decrease in the number of activated microglia. Similarly, a study in a rat model of Parkinson’s disease showed that tacrolimus did not reduce microglial activation but did reduce cell death [21]. Studies in adult rodents have shown that tacrolimus administration following middle cerebral artery occlusion decreased the number of microglia and reduced T-cell infiltration in the infarct area [22]. Additionally, another study showed that tacrolimus reduced macrophage recruitment in the brain following focal injury [18]. This suggests that the inhibitory effect tacrolimus has on the development of neonatal brain injury is unlikely due to the direct effects of microglia, indicating that T-cells may play a more prominent role in the development of neonatal brain injury than previously thought. When we administered tacrolimus post-HI injury, we found that only high doses of tacrolimus (>0.1 mg/kg/day) could mitigate brain injury, and this was coupled with a significant decrease in microglial activation, suggesting that immunomodulation is a key mechanism by which tacrolimus prevents HI brain injury when administered after injury is initiated.

Whilst previous studies in adult rodents with ischaemic brain injury have shown that infarct volume is decreased with tacrolimus administration [23,24,25,26,27], our study is the first to examine its effects on tissue loss in the neonatal brain. We found that pre-treatment with tacrolimus significantly reduced brain tissue loss, and tacrolimus administration post-HI injury at the highest doses tested was also effective. In addition, we observed a profound reduction in apoptotic cell death in animals that were administered tacrolimus before the onset of HI brain injury, which is consistent with adult rodent studies of ischaemic injury [28], indicating that tacrolimus may act on apoptotic pathways as a mechanism of action. Tacrolimus inhibits calcineurin, which plays a role in neuronal apoptosis [29,30], and may cause inhibition of nitric oxide synthase activity, decreasing nitric oxide production via the calcineurin pathway [31]. This is postulated to be a potential neuroprotective treatment strategy [32]. Furthermore, previous studies in mice have shown that a single intravenous injection of tacrolimus produced a rapid decrease in body temperature [33]. The authors suggest that hypothermia was likely induced by tacrolimus interacting with the CNS. As such, the neuroprotective mechanism of action of tacrolimus may involve hypothermia induction, although this requires further investigation.

Our study highlights that the timing and dose of immunomodulatory interventions to treat HI brain injury are critical for achieving a neuroprotective benefit. In this study, we performed a dose escalation of tacrolimus in neonatal rats at 0.025, 0.05, 0.1 and 0.25 mg/kg/day, which was informed by adult rodent studies of ischaemic brain injury, where the minimum effective dose to reduce brain volume loss was 0.1 mg/kg/day [27]. In our neonatal studies, the administration of tacrolimus after the onset of HI injury was only effective at higher doses, suggesting that there may be a narrow therapeutic window. Further adding to the challenge of treating HI brain injury with tacrolimus, our results showed that higher doses of tacrolimus (0.5 and 1.0 mg/kg/day) significantly decreased rat pup survival. The dose of 1 mg/kg/day resulted in a 60% mortality rate. Studies in both neonatal and adult rodents have used comparably high doses. Of particular note, a previous neonatal rodent study only administered tacrolimus at 1 mg/kg, for four days [18], while a study undertaken in adult rodents found that the most effective neuroprotective dose was 1 mg/kg [34]. However, given the high mortality rate we observed at this dose, tacrolimus appears to be more toxic to neonatal rodents. Our study, therefore, highlights the importance of long-term safety studies for potential therapeutic targets, particularly for perinatal brain injury.

Clinically, tacrolimus induces neurotoxicity in 10–28% of patients [35] which is thought to be due to endothelial damage and the subsequent release of endothelial toxins, causing seizures and tremors [36]. Other side effects include nephrotoxicity, glucose intolerance and hypertension [37]. Therefore, the clinical use of tacrolimus needs to be approached with caution, especially in neonates. Our study demonstrates that toxicity is evident when tacrolimus is administered at high doses. Although we found that administering tacrolimus prior to HI was superior to administering it post-HI, the optimisation of tacrolimus treatment after HI is critical to reduce the risk of side effects. This is consistent with the current clinical literature. In humans, tacrolimus is not a potential prophylactic treatment for neonatal HI brain injury due to known deleterious side effects. Finding different analogues or therapeutics that can inhibit the T-cell and/or calcineurin pathway but have fewer side effects than tacrolimus, or agents that could be given as a prophylactic, may prove beneficial as a treatment for neonatal HI brain injury. In addition, an unexpected finding was that IL-1β was significantly increased when tacrolimus was administered after brain injury. This may be of concern since this pro-inflammatory cytokine is strongly associated with neonatal brain injury [38,39,40]. The reason for the increase in IL-1β density remains unclear, especially since increased IL-1β is usually coupled with an Increase in microglial activation [41] which was not observed in this study and warrants further investigation.

A limitation of this study is that we did not investigate the overall number of peripheral immune cells, in particular, T-cells or their phenotype, via flow cytometry, which could provide more insight into the effects of tacrolimus on the interaction between localised neuroinflammation and the periphery. However, our findings suggest that T-cells and the IL-2 pathway could be involved in the progression of HI brain injury in the first 24 h after insult. In this study, we also did not perform behavioural testing to determine whether the neuroprotective effects we observed translated to improvement in motor and cognitive behaviours; obtaining this information is an important next step. Moreover, this study only investigated tacrolimus administration commencing at one time point after HI and, moving forward, it will be important to investigate the effect of initiating tacrolimus administration, and/or T-cell inhibition, at other timepoints following HI. It will also be necessary to investigate the effect of delayed tacrolimus administration in older pups with established brain injury. It is known that chronic inflammation is associated with adverse neurological outcomes [42], so identifying treatments that can suppress the immune system months after injury may also be beneficial. It will be necessary to undertake long-term studies to determine whether tacrolimus has a lasting effect on reducing perinatal brain injury.

## 5. Conclusions

This study has shown that tacrolimus is neuroprotective when administered prior to neonatal HI brain injury. These studies demonstrate the important role that T-cells and modulation of the immune system play in the initiation and development of neonatal HI brain injury. We have also shown that tacrolimus appears to work via the modulation of cell death pathways to prevent neonatal brain injury. In addition, we have demonstrated that high doses of tacrolimus are toxic in neonatal rat pups, and therefore, finding the right balance between an efficacious and safe dose is a major hurdle for the translation of tacrolimus as a treatment for neonatal HI brain injury. These studies highlight the potential of targeting similar neuroinflammatory pathways to treat neonatal brain injury, and further research into the role and targeted modulation of T-cells is warranted.

## Figures and Tables

**Figure 1 cells-12-02659-f001:**
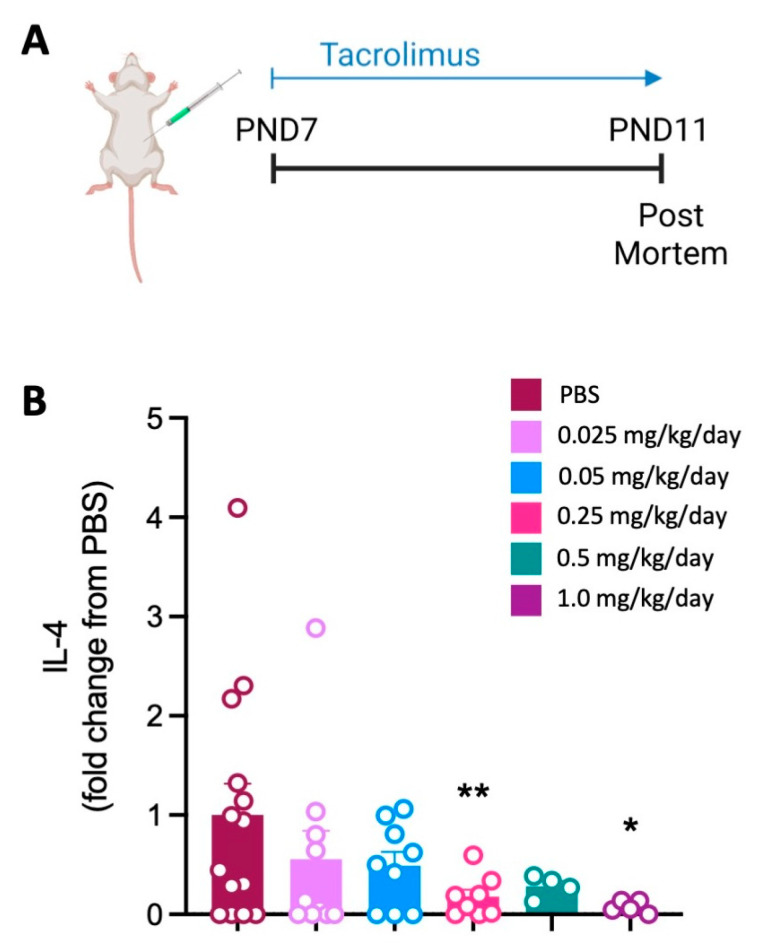
Spleen cytokine response is reduced within 3 days of initiating tacrolimus. (**A**) Experimental timeline. (**B**) Concentration of IL-4 protein produced by ionomycin and PMA-stimulated spleen cells. Data expressed as mean ± SEM; n = 4–14 animals/group; one-way ANOVA with Dunnett’s post hoc: * *p* < 0.05, ** *p* < 0.01 compared to PBS group.

**Figure 2 cells-12-02659-f002:**
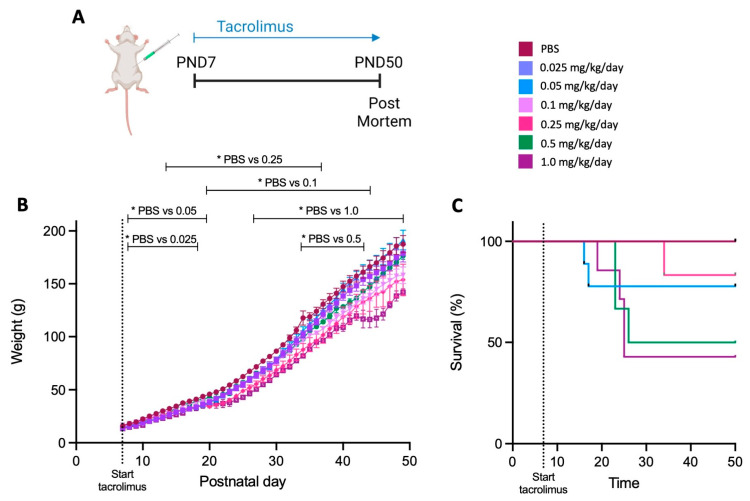
Higher doses of tacrolimus reduce body weight and survival. (**A**) Experimental timeline. (**B**) Rat pup body weight and (**C**) rat pup survival following intraperitoneal PBS (n = 16) or tacrolimus administration 0.025 mg/kg/day (n = 8), 0.05 mg/kg/day (n = 9), 0.1 mg/kg/day (n = 7), 0.25 mg/kg/day (n = 6), 0.5 mg/kg/day (n = 6) and 1.0 mg/kg/day (n = 7) starting at postnatal day seven until post-mortem at postnatal day 50. Data expressed as mean ± SEM; one-way ANOVA with Dunnett’s post hoc: * *p* < 0.05 compared to PBS group.

**Figure 3 cells-12-02659-f003:**
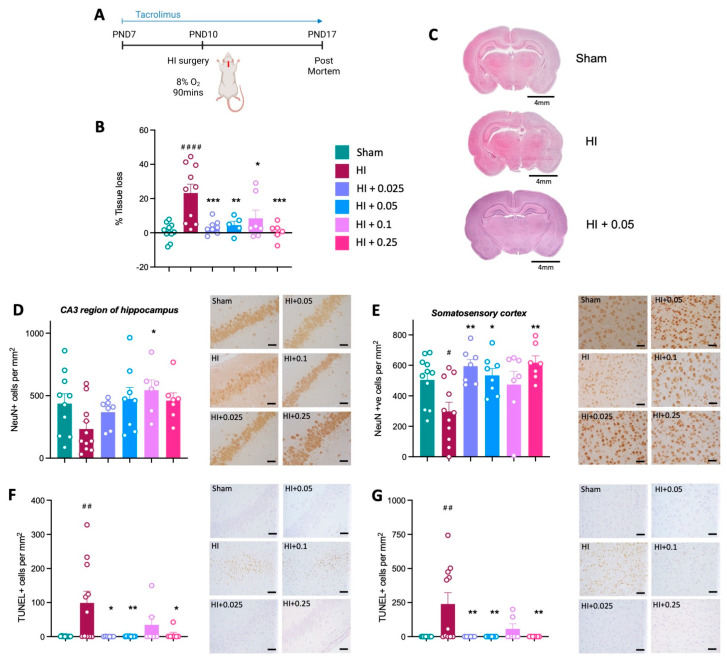
Pre-treatment with tacrolimus before HI brain injury prevents brain tissue loss and neuronal loss in the somatosensory cortex and reduces apoptosis. (**A**) Experimental timeline for tacrolimus pre-treatment. (**B**) Left hemisphere tissue loss as percentage of total brain area. (**C**) Representative images of haematoxylin and eosin staining showing brain tissue loss for sham, HI and HI + 0.05 mg/kg/day groups; scale bars = 4 mm. Number of NeuN-positive cells indicating neurons in the CA3 region of the hippocampus (**D**) and somatosensory cortex (**E**). Number of TUNEL-positive cells in the CA3 region of the hippocampus, # is where there is significance compared to sham group. (**F**) and somatosensory cortex (**G**). Data expressed as mean ± SEM; n = 5–11 animals/group; one-way ANOVA with Dunnett’s post hoc: #### *p* < 0.0001, ## *p* < 0.01 compared to sham group; * *p* < 0.05, ** *p* < 0.01, *** *p*<0.001, compared to HI control group. For representative immunohistochemistry images, scale bars = 50 μm.

**Figure 4 cells-12-02659-f004:**
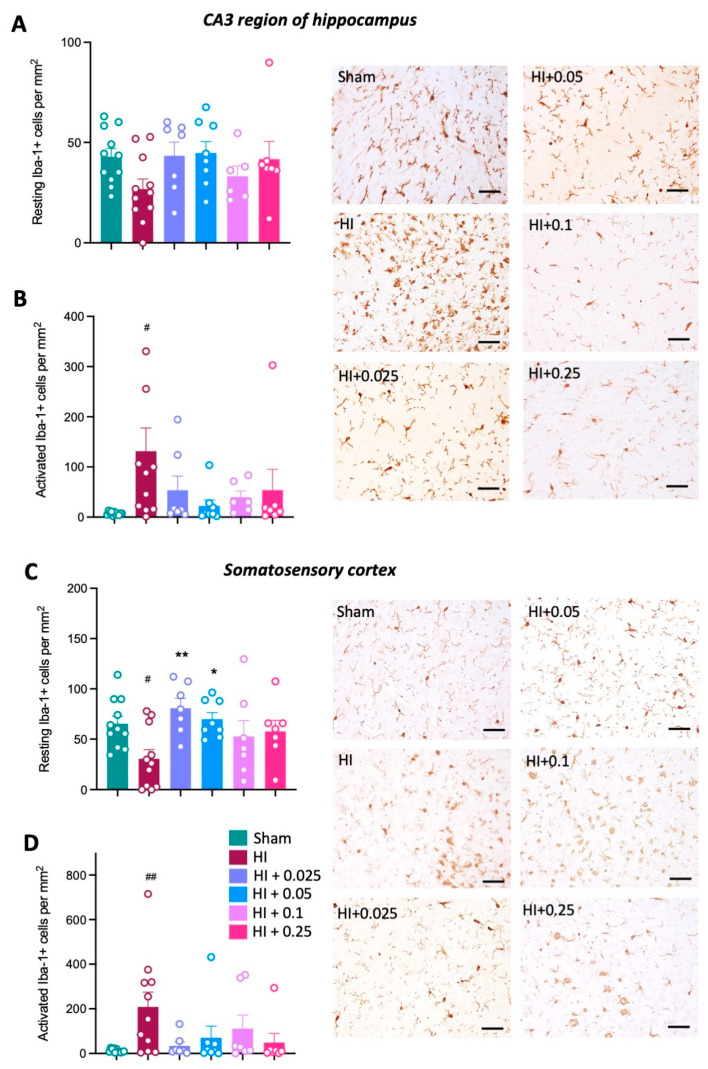
Effect of tacrolimus pre-treatment on microglial activation. Number of resting microglia cells in the CA3 region of the hippocampus (**A**) and somatosensory cortex (**C**). Number of activated microglia cells in the CA3 region of the hippocampus (**B**) and somatosensory cortex (**D**). Data expressed as mean ± SEM; n = 6–11 animals/group; one-way ANOVA with Dunnett’s post hoc: # *p* < 0.05, ## *p* < 0.01 compared to sham group; * *p* < 0.05, ** *p* < 0.01 compared to HI control group. Scale bars = 50 μm.

**Figure 5 cells-12-02659-f005:**
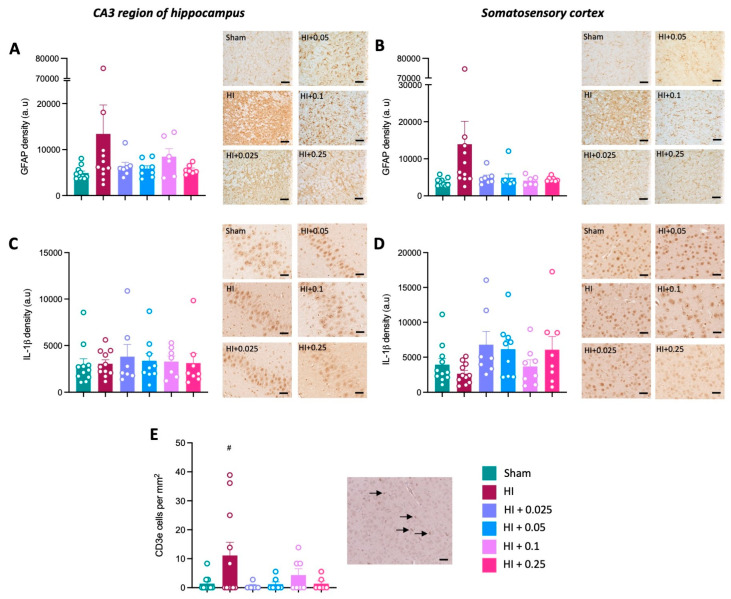
Effect of tacrolimus pre-treatment on neuroinflammation. GFAP density in the CA3 region of the hippocampus (**A**) and somatosensory cortex (**B**). IL-1β density in the CA3 region of the hippocampus (**C**) and somatosensory cortex (**D**). (**E**) Number of CD3e T-cells present in the left hemisphere, and representative image of CD3e staining in CA3 region of the hippocampus of an HI brain; the black arrows indicate CD3e-positive cells. Data expressed as mean ± SEM; n = 6–11 animals/group; one-way ANOVA with Dunnett’s post hoc: # *p* < 0.05, compared to sham group. Scale bars = 50 μm.

**Figure 6 cells-12-02659-f006:**
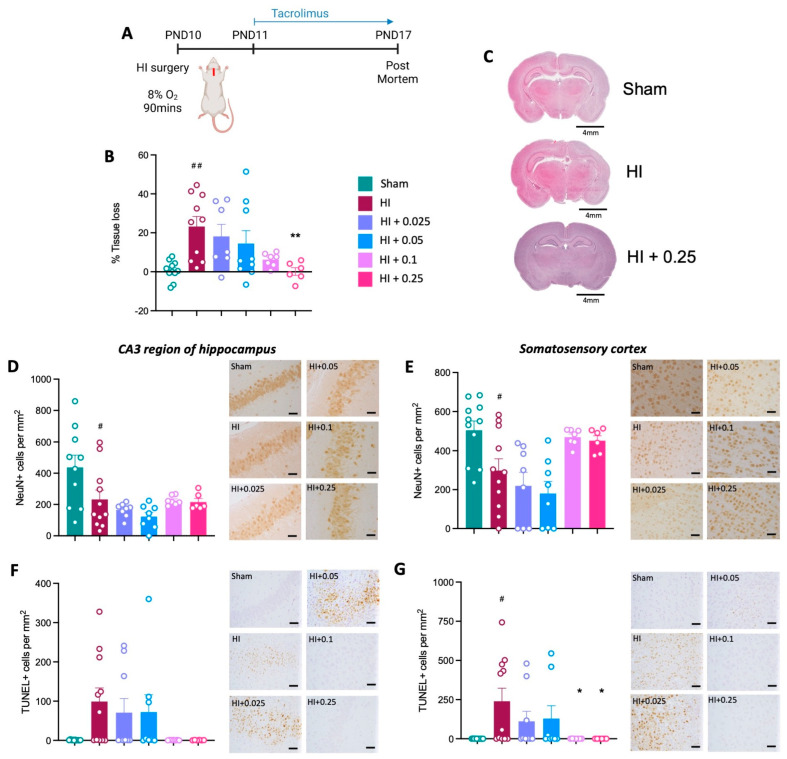
High doses of tacrolimus post-HI brain injury prevent brain tissue loss and neuronal cell loss in the somatosensory cortex. (**A**) Experimental timeline for tacrolimus post-HI treatment. (**B**) Left hemisphere tissue loss as a percentage of total brain area. (**C**) Representative images of haematoxylin-and-eosin-stained brain sections for brain tissue loss for sham, HI and HI + 0.25 mg/kg/day groups; scale bars = 4 mm. (**D**) Number of NeuN-positive cells in the CA3 region of the hippocampus and (**E**) somatosensory cortex. (**F**) Number of TUNEL-positive cells in the CA3 region of the hippocampus and (**G**) somatosensory cortex. Data expressed as mean ± SEM; n = 6–11 animals/group; one-way ANOVA with Dunnett’s post hoc: # *p* < 0.05, ## *p* < 0.01 compared to sham group; * *p* < 0.05, ** *p* < 0.01 compared to HI control group. For representative immunohistochemistry images, scale bars = 50 μm.

**Figure 7 cells-12-02659-f007:**
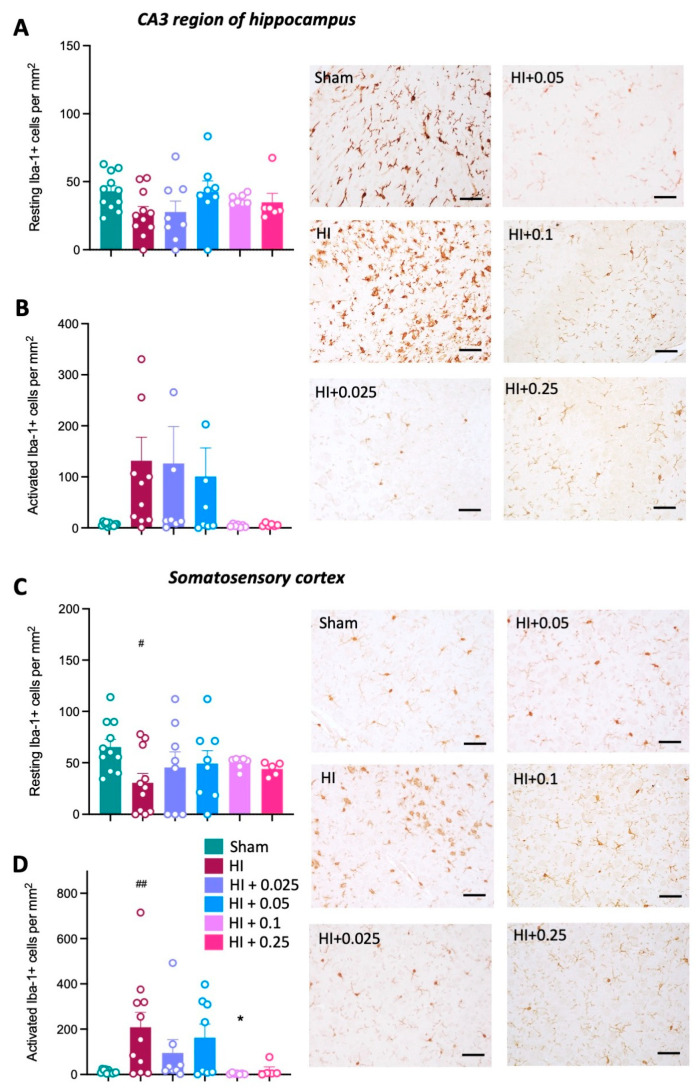
Effect of tacrolimus post-treatment on microglial activation. (**A**) Number of resting Iba-1-positive cells in the CA3 region of the hippocampus and (**C**) somatosensory cortex. Number of activated Iba-1-positive cells in the (**B**) CA3 region of the hippocampus and (**D**) somatosensory cortex. Data expressed as mean ± SEM; n = 5–11 animals/group; one-way ANOVA with Dunnett’s post hoc: # *p* < 0.05, ## *p* < 0.01 compared to sham group; * *p* < 0.05 compared to HI group. Scale bars = 50 μm.

**Figure 8 cells-12-02659-f008:**
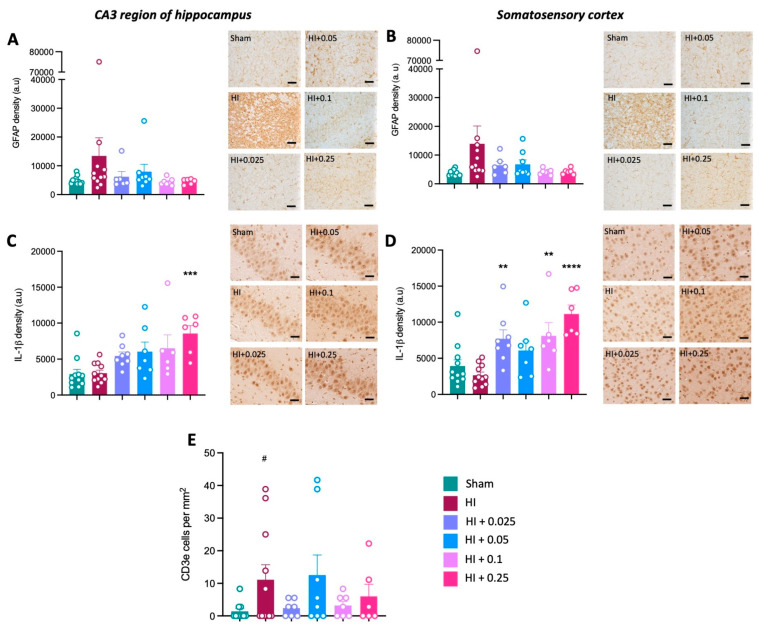
Effect of tacrolimus post-treatment on neuroinflammation. (**A**) GFAP density in the CA3 region of the hippocampus and (**B**) somatosensory cortex. IL-1β density in the (**C**) CA3 region of the hippocampus and (**D**) somatosensory cortex. (**E**) Number of CD3e T-cells present in the left hemisphere. Data expressed as mean ± SEM; n = 5–11 animals/group; one-way ANOVA with Dunnett’s post hoc: # *p* < 0.05 compared to sham group; ** *p* < 0.01, *** *p* < 0.001, **** *p* < 0.0001 compared to HI group. Scale bars = 50 μm.

## Data Availability

The data presented in this study are available on request from the corresponding author.

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
