# Peer review of "Neuroprotective Action of Tacrolimus before and after Onset of Neonatal Hypoxic–Ischaemic Brain Injury in Rats"

_cells, 2023, doi:10.3390/cells12222659_

Round 1

Reviewer 1 Report

Comments and Suggestions for Authors

This paper assessed the efficacy of TACROLIMUS in a neonatal rat model of Hypoxic-Ischemic Brain Injury. The article is well-crafted and presents neuroprotective strategies (various doses and age-related outcomes) for neonatal brain injuries.

Major Comments:

1. One key experiment appears to be missing: Immunohistochemistry on the brain at P50. According to Figure 2, these animals were generated, and it would be valuable for a long-term evaluation of tacrolimus's effects on neonatal brain injuries. Additionally, there's a concern about long-term weight loss and potential cognitive deficits. Even though cognitive assessment is not included in this paper, the discussion should address this issue. In the field, it's well-known that histological improvements don't necessarily equate to normal brain function after neuroprotective treatments for neonatal brain injuries.

Minor Comments:

1. Lines 185 and 190: The evaluation of the spleen does not constitute systemic evaluation. Plasma cytokine levels are missing; please replace with "spleen inflammation."

2. Add scale bars to all representative images.

3. In the figure legends, include information on the statistical tests applied and the sample size (n), as shown in Figure 2.

4. In Figure 4, the representative images suggest a significant reduction in IBA1+ cells in groups HI+0.1 and HI+0.25. The contrast also hints at labeling issues. Consider using alternative images. The same observation applies to the cortex groups SHAM, HI, HI+0.1, and HI+0.25. Make the representative images larger.

5. Standardize the use of capitalization for the "SHAM" group, both in the text and the figures.

6. In Figure 5E, explain why you're only looking at CA3 and not the cortex.

7. In the acknowledgments section, acknowledge the use of BioRender for designing some of the figures.

Author Response

Major Comments:

  1. One key experiment appears to be missing: Immunohistochemistry on the brain at P50. According to Figure 2, these animals were generated, and it would be valuable for a long-term evaluation of tacrolimus's effects on neonatal brain injuries. Additionally, there's a concern about long-term weight loss and potential cognitive deficits. Even though cognitive assessment is not included in this paper, the discussion should address this issue. In the field, it's well-known that histological improvements don't necessarily equate to normal brain function after neuroprotective treatments for neonatal brain injuries.

We agree that the long-term effects of tacrolimus on the neonatal brain is valuable information, and we did actually perform these studies (behavioural testing (novel object recognition) and neuron cell counts (NeuN) on a subset of these long term animals), but did not include the data in the original manuscript as we found no effect.

We found no difference between any groups in the brain weight at post mortem at PND50 (Suppl. Fig 1A), or the novel object recognition test at PND50 (Suppl. Fig 1B. We also assessed neuron cell counts in the CA3 region of the hippocampus at PND50 in a small cohort of animals that were given the highest doses of tacrolimus (0.5 and 1.0mg/kg/day) and compared to PBS animals and we found no difference (Suppl. Fig 1C).

We have now added a Supplementary figure 1 which includes this data and we have amended the text in the results section (lines 219-224).

We also agree with your point that histological improvements do not always equate to improved brain function and cognition. We agree that the lack of behavioural data in this study is a limitation and have included this point in our discussion on (lines 477-479).

Minor Comments:

  1. Lines 185 and 190: The evaluation of the spleen does not constitute systemic evaluation. Plasma cytokine levels are missing; please replace with "spleen inflammation."

We have changed ‘systemic inflammation’ to ‘spleen inflammation’ in lines 199 and 205.

  1. Add scale bars to all representative images.

We have added scale bars to all the representative images, each scale bar represents 50mm

  1. In the figure legends, include information on the statistical tests applied and the sample size (n), as shown in Figure 2.

We have added information into every figure legend about the statistical test used. We have also made sure all figure legends have the n number range and all the graphs include individual dot points for each n value. Given there are so many groups for each figure, we have a detailed animal number table in Supplementary Table 1-3 that can be referred to for exact numbers used for each group, if required.

  1. In Figure 4, the representative images suggest a significant reduction in IBA1+ cells in groups HI+0.1 and HI+0.25. The contrast also hints at labeling issues. Consider using alternative images. The same observation applies to the cortex groups SHAM, HI, HI+0.1, and HI+0.25. Make the representative images larger.

For all Iba-1 analysis, cell counts were performed where manual counting was conducted (not densitometry), so the contrast difference will not have affected the cell count analysis. We have used more appropriately white balanced representative images that have similar contrast to each other and made the representative images for Iba-1 analysis in Figure 4 and 7 larger.

  1. Standardize the use of capitalization for the "SHAM" group, both in the text and the figures.

We have amended and standardised all use of Sham throughout the text and figures.

  1. In Figure 5E, explain why you're only looking at CA3 and not the cortex.

In this stain assessing T-cells (using anti CD3e) the entire left hemisphere was analysed, including both the CA3 and the cortex. The reason we chose to assess the whole left hemisphere and not individual regions is because the number of T-cells in the brain is low and it is hard to get accurate counts when just assessing one small individual region. The CA3 region was included as a representative image in this instance, we have amended the text to make this clearer.

  1. In the acknowledgments section, acknowledge the use of BioRender for designing some of the figures.

We have included an acknowledgement of Biorender in the acknowledgement section, line 520-521.

Reviewer 2 Report

Comments and Suggestions for Authors

This study examined the neuroprotective actions of treatment with the immunosuppressant tacrolimus either initiated prior to and following or just following hypoxia-ischemia in neonatal rats. The study is well performed and well written, the methods and model are suitable, and the study provides very interesting results, and suggests that targeting neuroinflammation in the form of T-cells may be a potential therapeutic approach to reducing brain injury.

I have a number of comments/questions below:

Please provide some discussion on the use of pretreatment as a strategy for reducing brain injury, and any potential limitations of this approach. For example, there is potential for pretreatment to reduce the severity of the insult itself, and thus any protection may relate, at least in part, to a lesser insult rather than a direct action on neuroinflammation. Also, please comment on the potential for the drug to induce hypothermia, and whether this may affect results. For example,  see ‘A tacrolimus-related immunosuppressant with reduced toxicity. F J Dumont 1, S Koprak, M J Staruch, A Talento, G Koo, C DaSilva, P J Sinclair, F Wong, J Woods, J Barker, J Pivnichny, I Singer, N H Sigal, A R Williamson, W H Parsons, M Wyvratt’.

Were there any effects of sex on the outcomes?

Please consider providing overall ANOVA data in the figure legends so that the reader can see the overall effect, before post-hoc analysis.

I may have missed this, but were there any mortality data for the drug + HI groups, rather than just drug alone.

Other

Overall, I felt the methods could be slightly tidied up for readability. For example:

In the methods, please provide details on whether injections were performed on the same time of the day for each animals/litter (the same for HI).

Please add animal numbers in the various groups and the sex distribution into the methods.

In the histology and immuno sections, can you please provide more details of magnification used, the numbers of sections used, and the levels of the brain used. Were all analyses performed in the SS Ctx and CA3 regions.

Please clarify what antibodies/measures were assessed by cell counts vs densitometry, and how the data are presented (e./g. mm2 vs XXX), and what regions were assessed?

I was a little confused on the brain tissue loss analysis, as the methods state ‘For percentage tissue loss, the difference in area between the contralateral and ipsilateral hemispheres over the contralateral hemisphere area was calculated as: (area of the contra-lateral hemisphere – area of the ipsilateral hemisphere)/(area of the contralateral hemisphere)’ while in the text  for Fig. 3 it states ‘Brain injury was confirmed following HI, as evidenced by significant left hemisphere tissue loss in the HI control group compared to sham control animals’. I.e., there is no mention of sham controls being used in the methods for %tissue loss.

It was a little unclear in the methods how/and for what the sham surgery animals were used. Were they part of the neuropathological assessment? Please clarify.

In the immuno section, please clarify how many nights/time the primaries were added for, and the timing for the secondaries.

Was the assessment of CD3 cells done differently to the other cells/stains? This is a little unclear as it was stated at start of immuno section that  the SS Ctx and CA3 regions were assessed, but later it stated that ‘To quantify the number of CD3 positive cells, the entire left hemisphere was imaged and cells were counted.’

What hemispheres/groups were compared for the various cell counts with injury/treatment? Was in Ipsilateral vs contralateral, or were sham animals used?

Author Response

The manuscript is well-written, coherent, and well-explained. Methods and data were presented very clearly and the authors described their findings in the context of what is known and what needs to be known very nicely. I did not find any technical issues. However, I found the following minor formatting issues that need to be corrected before publishing;

  • In the titles of all the figures, it is mentioned that "mean + SEM" However, It should be mean ± SEM

Thank you, this error has been amended in all figure legends.

  • The data availability statement, Appendix A, and Appendix B are copies of the general format. So, either remove it or write it according to the manuscript's need.

The data availability statement has been amended and the Appendices have been removed.

Reviewer 3 Report

Comments and Suggestions for Authors

The manuscript is well-written, coherent, and well-explained. Methods and data were presented very clearly and the authors described their findings in the context of what is known and what needs to be known very nicely. I did not find any technical issues. However, I found the following minor formatting issues that need to be corrected before publishing;

1- In the titles of all the figures, it is mentioned that "mean + SEM" However, It should be mean ± SEM

2- The data availability statement, Appendix A, and Appendix B are copies of the general format. So, either remove it or write it according to the manuscript's need.

Author Response

Please provide some discussion on the use of pretreatment as a strategy for reducing brain injury, and any potential limitations of this approach. For example, there is potential for pretreatment to reduce the severity of the insult itself, and thus any protection may relate, at least in part, to a lesser insult rather than a direct action on neuroinflammation.

We agree with your comment and we do not postulate that pre-treatment with tacrolimus would be an effective strategy for reducing brain injury, as most of the time you do not know which babies are going to develop injury to administer a prophylactic treatment. The reason for including the pre-treatment in this study was to investigate the effect of depleting T cells via tacrolimus administration. As stated in our introduction (line 68-69)- pre-treatment was to “ascertain the effect of suppressing the immune system on the initiation and development of injury”. We have included additional text in the discussion to emphasise this point (line 393-395; 407-410).

Also, please comment on the potential for the drug to induce hypothermia, and whether this may affect results. For example, see ‘A tacrolimus-related immunosuppressant with reduced toxicity. F J Dumont 1, S Koprak, M J Staruch, A Talento, G Koo, C DaSilva, P J Sinclair, F Wong, J Woods, J Barker, J Pivnichny, I Singer, N H Sigal, A R Williamson, W H Parsons, M Wyvratt’.

We have added comment on the potential of tacrolimus to reduce hypothermia in the discussion in lines 427-431.

Were there any effects of sex on the outcomes?

We do not have a large enough n number in this study, which means we are underpowered to determine the effect of sex on each of these outcomes. We did ensure that the split of males and females was as even as possible in each of our groups and we have included the breakdown of males and females used for each treatment group in Supp Tables 1-3.

Please consider providing overall ANOVA data in the figure legends so that the reader can see the overall effect, before post-hoc analysis.

Anytime we have provided individual post-hoc analysis significance in the graphs, the overall ANOVA result was significant. Given the amount of data presented in each figure adding the extra information makes the figures very messy. Instead, we have amended the text in the statistics section to state that when post hoc significance is presented, the overall ANOVA was significant (line 194-195).

Overall, I felt the methods could be slightly tidied up for readability. For example: In the methods, please provide details on whether injections were performed on the same time of the day for each animals/litter (the same for HI).

The injections were conducted in a within two hour window each day, in the morning. We have added this information to the methods section in lines 110-111.

Please add animal numbers in the various groups and the sex distribution into the methods.

We have all of the animal numbers for each treatment group included in a supplementary file (Supp Table 1-3). We have also added the sex distribution for each treatment group into these tables.

In the histology and immuno sections, can you please provide more details of magnification used, the numbers of sections used, and the levels of the brain used. Were all analyses performed in the SS Ctx and CA3 regions.

We have provided more details in the method section to make the details of the immunohistological experiments clearer. We performed all analysis at 400X magnification. Duplicate brain sections were used for each analysis, and for all immunohistochemistry analysis three fields of view per duplicate section were analysed for each brain region.

We assessed the brain at approximately -4.2mm bregma as these coronal sections contain both somatosensory cortex and the hippocampus. Yes all analysis were performed for the somatosensory cortex and the hippocampus except where described in the text, i.e T cell counts (CD3e) was performed across the entire left hemisphere as the numbers were too low to specifically assess the somatosensory cortex or the hippocampus alone.

Please clarify what antibodies/measures were assessed by cell counts vs densitometry, and how the data are presented (e./g. mm2 vs XXX), and what regions were assessed?

We have clarified which antibodies were assessed by cell counts vs densitometry. Data was presented as mm2 for cell counts and as density (a.u.) for densitometry analysis as seen in figure legends. Please see the comment above for the relevant brain regions.

I was a little confused on the brain tissue loss analysis, as the methods state ‘For percentage tissue loss, the difference in area between the contralateral and ipsilateral hemispheres over the contralateral hemisphere area was calculated as: (area of the contra-lateral hemisphere – area of the ipsilateral hemisphere)/(area of the contralateral hemisphere)’ while in the text  for Fig. 3 it states ‘Brain injury was confirmed following HI, as evidenced by significant left hemisphere tissue loss in the HI control group compared to sham control animals’. I.e., there is no mention of sham controls being used in the methods for %tissue loss.

To analyse tissue loss it is a calculation that compares the injured ipsilateral hemisphere to the non-injured contralateral hemisphere, this is a fairly standard method for this kind of assessment. This calculation is performed individually for every animal in each group. There is no comparison to sham in the individual tissue loss calculation. We only make the comparison using the averaged group data after individual calculations, ie sham have ~0% tissue loss vs HI have ~20% tissue loss. We have tried to make this clearer (lines 152-154).

It was a little unclear in the methods how/and for what the sham surgery animals were used. Were they part of the neuropathological assessment? Please clarify.

The sham group makes up one of our important control groups for this experiment that controls for the effect of surgery without HI induction, so that group was included in all analysis including the immunohistochemistry and tissue loss analysis. We have made in use of this sham group clearer in the methods section (lines 106, 126, 146).

In the immuno section, please clarify how many nights/time the primaries were added for, and the timing for the secondaries.

The information for the length of incubation has been added to 2.8 immunohistochemistry, line 166-167.

Was the assessment of CD3 cells done differently to the other cells/stains? This is a little unclear as it was stated at start of immuno section that  the SS Ctx and CA3 regions were assessed, but later it stated that ‘To quantify the number of CD3 positive cells, the entire left hemisphere was imaged and cells were counted.’

We have added clarification in line 179-182 that the CA3 and Ctx regions were analysed for Iba-1, GFAP, TUNNEl, IL-1b and NeuN. For CD3 positive cells, the entire left hemisphere was analysed.

What hemispheres/groups were compared for the various cell counts with injury/treatment? Was in Ipsilateral vs contralateral, or were sham animals used?

For all immunohistochemical analysis we always used the brain regions in the ipsilateral (left injured) hemisphere (clarification added line 159). To assess injury and improvement we always compared to sham animals, not to the contralateral hemisphere. While there is likely to be limited injury in the contralateral hemisphere, this model (rice vannuci HI) does induce systemic inflammation and therefore there may be a mild effect in the contralateral hemisphere and therefore this is not a reliable comparison.

The only assessment that used the contralateral hemisphere is the tissue loss, but for our injury comparison we still compare to sham animals as described in more detail above.

Round 2

Reviewer 1 Report

Comments and Suggestions for Authors

Scale bars are not in ALL representative images...

Author Response

We apologise that we missed the addition of scale bars to the full coronal brain sections shown in Figure 3C and Figure 6C. We have now added scale bars to those images.